# VIDARC: EMBODIED VIDEO DIFFUSION MODEL FOR CLOSED-LOOP CONTROL

## ABSTRACT

Robotic arm manipulation in data-scarce settings is a highly challenging task due to the complex embodiment dynamics and diverse contexts. Recent video-based approaches have shown great promise in capturing and transferring the temporal and physical interactions by pre-training on Internet-scale video data. However, such methods are often not optimized for the embodiment-specific closed-loop control, typically suffering from high latency and insufficient grounding. In this paper, we present Vidarc (Video Diffusion for Action Reasoning and Closed-loop Control), a novel autoregressive embodied video diffusion approach augmented by a masked inverse dynamics model. By grounding video predictions with action-relevant masks and incorporating real-time feedback through cached autoregressive generation, Vidarc achieves fast, accurate closed-loop control. Pre-trained on one million cross-embodiment episodes, Vidarc surpasses state-of-the-art baselines, achieving at least a 15% higher success rate in real-world deployment and a 91% reduction in latency. We also highlight its robust generalization and error correction capabilities across previously unseen robotic platforms.

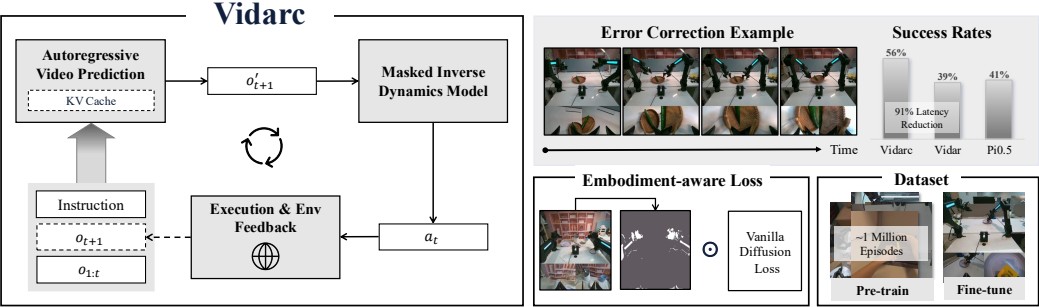

Figure 1: Left: Vidarc consists of an embodied autoregressive video diffusion model and a masked inverse dynamics model. To enable closed-loop control, the inference pipeline re-prefills environment feedback into the autoregressive video generation. Right: After being pre-trained on approximately one million bimanual demonstration episodes, Vidarc is fine-tuned on an unseen platform using calibration with embodiment-specific masks; it achieves state-of-the-art performance and exhibits robust error correction capabilities.

## 1 INTRODUCTION

Robotic arm manipulation is a fundamental yet higly complex task, requiring precise coordination across multiple degrees of freedom to execute intricate movements in dynamic environments. In many real-world applications, such as autonomous assembly lines, medical surgery, or hazardous material handling, collecting large, high-quality datasets is prohibitively expensive or impractical, especially when adapting robotic control to new platforms, tasks, or environments. As a result, achieving robust and generalizable manipulation skills from limited data is a crucial goal, enabling widespread and scalable deployment of robotic systems (Kroemer et al., 2021; Yang et al., 2025).

Inspired by the success of large language models, one effective approach in data-scarce settings is to leverage a pre-trained foundation model plus a fine-tuning step for knowledge transfer. Representative progress includes meta-learning (Finn et al., 2017), vision-language-action models which typically add action heads to pre-trained vision-language models (Kim et al., 2024; Intelligence et al., 2025; Liu et al., 2024a; Song et al., 2025), as well as video generation models with lightweight embodied-specific controllers or inverse dynamics models (Feng et al., 2025; Liao et al., 2025). Among these approaches, video generation models have shown great promise by fully exploring the Internet-scale video data, while the others often have to collect a large set of human demonstration data. Videos, unlike static images or discrete trajectory representations, capture the full temporal dynamics and interaction cues essential for manipulation tasks. Trained on massive video datasets (Wang et al., 2025; Liu et al., 2024b; Kong et al., 2024), video generation models create transferable priors that enforce physical consistency, support counterfactual reasoning, and can be efficiently fine-tuned with very few demonstrations (Feng et al., 2025).

This progress notwithstanding, little progress has been made so far on the real-time, embodiment-specific requirements of robotic control. Closed-loop control is highly desired and especially important in robotics (Ye et al., 2025; Xue et al., 2025; Sun et al., 2024; Black et al., 2025) because it enables the system to constantly refine its actions based on new sensory feedback, greatly increasing robustness to unexpected environmental changes, errors, or perturbations. Achieving this with video foundation models poses unique challenges: it requires low-latency generation, seamless integration of real-time feedback, and quick adaptation to embodiment-specific cues within the video stream. Previous approaches often focused on open-loop prediction or required slow, sequential inference, making them impractical for real-world, interactive robot tasks (Du et al., 2023). Moreover, pure video generative models typically lack grounding in embodiment-relevant dynamics and visual features; subtle visual or physical deviations—such as minor changes in the robot arm's appearance or pose—can cause dramatic task failures if not properly accounted for (Zhao et al., 2022).

To address these limitations, we propose **Vidarc** (Video Diffusion for Action Reasoning and Closed-loop Control), which consists of an autoregressive embodied video diffusion model and a masked inverse dynamics model. By incorporating environmental feedback in the autoregressive generation process with key-value (KV) caching, Vidarc enables robust closed-loop control with low latency during inference. To further ground the video diffusion model in the specific dynamics of a robot, we use learned action-relevant masks from the masked inverse dynamics model to construct an embodiment-aware diffusion loss, ensuring the generated videos are actionable. We illustrate our method in Figure 1.

With large-scale cross-embodiment pre-training on approximately one million episodes, Vidarc adapts to an unseen real-world platform with superior success rates than strong baselines: 17% higher than Vidar (Feng et al., 2025) and 15% higher than Pi0.5 (Intelligence et al., 2025). Furthermore, Vidarc only incurs 8.8% of Vidar's latency, with remarkable generalization and error correction capabilities.

## 2 PREREQUISITE

We start by briefly summarizing the preliminary knowledge.

### 2.1 DIFFUSION MODEL

Diffusion model designs a noise injection and denoise process to generate high-quality images or videos. Modern video diffusion models (Wang et al., 2025; Xie et al., 2025) adopt the flow matching framework (Lipman et al., 2023; Liu et al., 2023), which enables stable training with the ordinary differential equation (ODE) formulation. Given a video $x_1$, a random Gaussian noise $x_0$ with the same size, and a timestep $t \in [0, 1]$, we define the noised video $x_t$ as $tx_1 + (1 - t)x_0$. Let $\mathcal{V}$ be the video space, and $\mathcal{C}$ be the condition space. The diffusion model learns a flow function $v_\theta : \mathcal{V} \times \mathbb{R} \times \mathcal{C} \to \mathcal{V}$, which parameterizes the vector field that transforms $x_1$ to $x_0$ given the intermediate $x_t$ and $t$ and condition $c$. The training objective is:

$$\mathcal{L}_{\text{diffusion}} = \mathbb{E}_{x_0,x_1,t,c} \left[ \|v_\theta(x_t, t, c) - (x_0 - x_1)\|_2^2 \right]. \tag{1}$$

During inference, we can sample from the learned distribution by solving the following ODE from $t = 0$ to $t = 1$ by using efficient training-free solvers Lu et al. (2022); Song et al. (2021):

$$\frac{\mathrm{d}x_t}{\mathrm{d}t} = v_\theta(x_t, t, c), \ t \in [0, 1]. \tag{2}$$

Thanks to their strong ability to model complex spatial-temporal dynamics, video diffusion models can serve not only as generative models but also as world models that simulate the evolution of visual environments. Recent works have demonstrated their potential for interactive prediction and control, such as physical simulation (Ball et al., 2025; He et al., 2025a) and robotic manipulation planning (Feng et al., 2025; Du et al., 2023). As both model capacity and training data scale up, video diffusion models exhibit emerging properties including zero-shot generalization and chain-of-frames reasoning (Wiedemer et al., 2025), suggesting promising applicability to complex real-world manipulation and reasoning tasks.

## 2.2 VIDEO-BASED ACTION PREDICTION

In video-based approaches, video diffusion models form the backbone, with actions derived either from an action head, which takes latent vectors as input (Hu et al., 2024), or from an Inverse Dynamics Model (IDM) (Tan et al., 2025), typically predicting actions $\hat{a}$ from images $x$. However, only the robotic arm's key regions are necessary for action prediction in these images, while other areas may introduce noise that interferes with model performance. To address this, Vidar (Feng et al., 2025) introduces a masked inverse dynamics model (MIDM) approach, which employs a mask predictor $U$ to predict a mask $m \in [0, 1]$ that highlights action-relevant pixels, together with an action regressor $R$ for action regression:

$$m = U(x), \quad \hat{a} = R(\text{Round}(m) \odot x), \tag{3}$$

where "Round" is the rounding function. This masking mechanism preserves critical motion-related regions and suppresses irrelevant visual information, thereby enhancing the accuracy and robustness of action prediction in the IDM. Since the mask is closely tied to the robot's dynamics, it offers a more effective prior than general segmentation models. In the training process, Vidar regularizes the area of $m$ with a weight of $\lambda$:

$$\mathcal{L}_{\text{action}} = \mathbb{E}_{x,a} \left[ l(\hat{a} - a) + \lambda \|m\|_1 \right], \tag{4}$$

where $l(.)$ is the Huber loss. Generally, the IDM handles robot-specific action spaces and control signals, while video diffusion models focus on unified video generation tasks, where abundant prior knowledge is transferred from pre-training.

## 3 METHOD

Although recent video world models have shown remarkable generalization across visual domains, their architectures are inherently not optimized for embodied control. Most of these methods rely on bidirectional diffusion mechanisms, which lack causality and suffer from high per-frame latency (Wang et al., 2024), global dependencies, and cumulative prediction errors in long-horizon sequences (Deng et al., 2025; Gao et al., 2025a; Huang et al., 2024; Yin et al., 2025). Moreover, the conventional diffusion training paradigm treats all visual features equally, ignoring the asymmetric, motion-dependent structure of physical interaction—a property essential for stable and efficient action generation.

In contrast, a robot-native policy must process environmental feedback in a causal and low-latency manner, enabling rapid perception–action cycles and continuous adaptation to dynamic surroundings. It should exhibit strong inductive biases for motion and kinematics, ensuring physically consistent trajectories, while maintaining computational efficiency to support real-time inference and robust generalization across diverse scenarios.

To meet these requirements, Vidarc builds upon causal autoregressive frame prediction with re-prefilling to enable closed-loop interaction with minimal latency. By re-prefilling observations from the environment, we can bridge the inherent training-inference gap in autoregressive models, preventing error accumulation and accounting for environmental changes. Further enhanced with an embodiment-aware loss, our model explicitly emphasizes motion dynamics during training, leading to more stable, efficient, and adaptive embodied behavior.

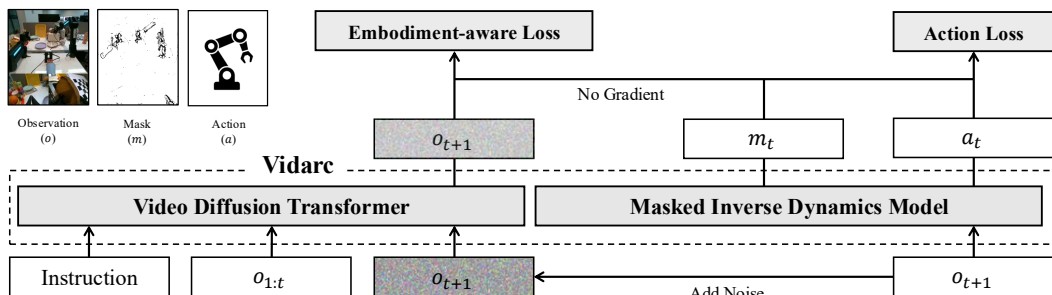

Figure 2: Vidarc comprises a video diffusion transformer and a masked inverse dynamics model. The video diffusion transformer is trained via teacher-forcing to predict the next observation based on previous observations and language instructions, while the masked inverse dynamics model is trained to infer actions from observations using a learnable masking mechanism that focuses attention on action-relevant regions. The learned mask is also used to reweight the diffusion loss, enhancing the video model's focus on regions important for action prediction.

## 3.1 MODEL DESIGN

Figure 3 presents an overview of our method. Specifically, let $\mathcal{L}$ be the language instruction space, $\mathcal{O}$ be the chunked visual observation space (a sequence of images with a chunk size $\geq 1$), and $\mathcal{A}$ be the chunked action space (a sequence of actions with a chunk size $\geq 1$). We target to learn a conditional robot manipulation policy $\pi : \mathcal{L} \times \mathcal{O} \to \mathbb{P}(\mathcal{A})$. Similar to previous video-based methods (Du et al., 2023), we decompose it into two models: $\pi = G \circ I$, where $G : \mathcal{L} \times \mathcal{O} \to \mathbb{P}(\mathcal{O})$ is a video generation model and $I : \mathcal{O} \to \mathcal{A}$ is an inverse dynamics model. The inverse dynamics model is typically modeled as a mapping from image to action; with a slight abuse of notation, we also use the term "inverse dynamics model" to refer to the batch application of image chunks as input.

To adopt closed-loop control, we propose to take the feedback from the environment into the pipeline. Assuming the transition function is $\mathcal{T} : \mathcal{O} \times \mathcal{A} \to \mathcal{O}$ and the observation aggregation function is $\mathcal{C} : \bigsqcup_{n=1}^{\infty} \mathcal{O}^n \to \mathcal{O}$, we can unroll the policy for timestep $t$ as follows:

$$\begin{cases} \hat{o}_{t+1} \sim G(l, \mathcal{C}(o_1, \cdots, o_t)) & \text{\# Autoregressive Generation} \\ a_t = I(\hat{o}_{t+1}) & \text{\# Inverse Dynamics Decoding} \\ o_{t+1} = \mathcal{T}(o_t, a_t) & \text{\# Execution and Collection,} \end{cases} \tag{5}$$

where $o_1$ is the initial observation. Specifically, we first generate the next observation $\hat{o}_t$ based on the instruction $l$ and the aggregation of previous observations. Then we decode the action $a_t$ from the generated observation $\hat{o}_t$. Finally, we execute the action $a_t$ in the environment and collect the new observation $o_{t+1}$.

## 3.2 TRAINING

We outline the training of the video diffusion transformer.

**Causal Training** To enable causal generation, we utilize the CausVid (Yin et al., 2025) method, transfer the text-image to video model to a frame-by-frame generation model. During the generation of each frame, all previous frames of this frame ($x_{prev}$) are noise-free (i.e., already denoised) and can be attended to in the attention operation. The causal training objective is:

$$\mathcal{L}_{\text{causal}} = \mathbb{E}_{x_0, x_1, t, c} \left[ \| v_\theta(x_t, t, c, x_{prev}) - (x_0 - x_1) \|_2^2 \right]. \tag{6}$$

**Embodiment-aware Loss** As shown in Figure 3.2, Video diffusion models suffer from inaccurate modeling of robot-relevant features, which are crucial for precise control. To address this issue, we propose an embodiment-aware loss that enhances the video diffusion model's focus on action-relevant regions. In this way, the model is adapted to the specific embodiment, enabling the generation of more actionable videos.

**Algorithm 1** Inference Algorithm of Vidarc

1: **Input:** Environment $E$, Instruction $l$, Autoregressive model $G$, Inverse Dynamics Model $I$
2: **Hyperparameters:** Chunk size $n\_c$, Maximum KV length $n\_k$
3: **while** task not complete **and** not timeout **do**
4:     $o_1 \leftarrow E.get\_obs()$
5:     $C = \{c_1\} \leftarrow G.prefill(o_1)$ *# Initialize the KV cache with the first observation*
6:     **for** $i = 1$ **to** $n\_k$ **step** $n\_c$ **do**
7:         $gen\_obs \leftarrow [\,]$
8:         **for** $j = i + 1$ **to** $i + n\_c$ **do**
9:             $o'_j, c'_j \leftarrow G.generate(l, C)$ *# Generate with KV cache*
10:            $gen\_obs.append(o'_j); C.append(c'_j)$
11:        **end for**
12:        $a \leftarrow I(gen\_obs)$
13:        $E.execute(a)$
14:        $C.pop\_back(n\_c)$ *# Remove the last chunk of the KV cache*
15:        $gt\_obs = \{o_{i+1}, ... o_{i+n\_c}\} \leftarrow E.get\_obs()$ *# Get ground truth observations*
16:        $C \leftarrow G.chunk\_prefill(C, gt\_obs)$ *# Re-prefill with ground truth observations*
17:    **end for**
18:    $C.clean()$ *# Reset the KV cache*
19: **end while**

Inspired by the masked inverse dynamic model, the mask $m$ highlights regions relevant to the robot's actions, such as the robot arm. We use the learned mask to reweight the diffusion loss, encouraging the video model to pay more attention to these critical areas. The final training objective is

$$\mathcal{L}_{\text{embodiment-aware}} = \mathbb{E}_{x_0, x_1, t, c} \left[ \|(1 + \eta \cdot U(x_1)) \odot (v_\theta(x_t, t, c, x_{prev}) - (x_0 - x_1))\|_2^2 \right], \quad (7)$$

where $\eta$ is a hyperparameter controlling the strength of the reweighting.

In this way, the video diffusion model is guided to focus on action-relevant regions, also matching the masked prediction of the inverse dynamics model (details in Section 2.2), leading to improved performance in precise control tasks.

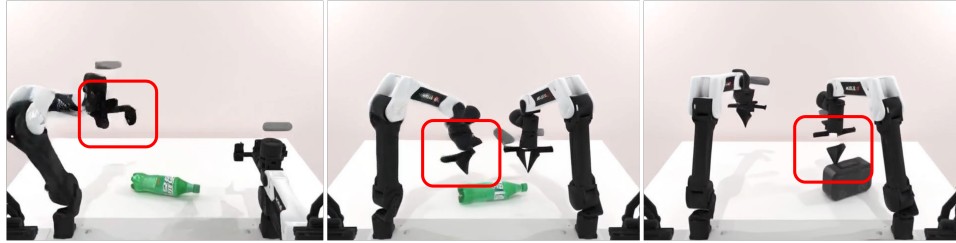

Figure 3: Video predictions often get artifacts around the robot arm, which affects the task success.

### 3.3 INFERENCE

The general inference pipeline is described by equation 5, with our detailed implementation provided in Algorithm 1. In particular, we generate the next observation based on previous real-world observations, rather than generated ones, enabling closed-loop control. This paradigm aligns with teacher forcing training, incorporating inference to prevent error accumulation.

To accelerate the generation process, we employ KV caching and cache instruction embeddings, thereby avoiding redundant recomputation across decoding steps. To further reduce inference latency, we introduce a re-prefill mechanism that optimizes the prefill phase: rather than recomputing KV caches for the entire sequence of prior observations, we only pop the latest generated KV cache, and then perform chunk prefill with the latest observations. In this way, the step sequence length for prefilling is significantly reduced, saving the computation cost and reducing the latency.

# 4 EXPERIMENTS

We now present experimental results with the goal to verify the following claims:

**C1:** Vidarc achieves superior success rates on both simulated and real domains;

**C2:** Vidarc generalizes effectively to unseen tasks and environments;

**C3:** Vidarc achieves low-latency closed-loop control with error correction abilities;

**C4:** The embodiment-aware diffusion loss enhances Vidarc's ability.

## 4.1 EXPERIMENTAL SETUP

**Hardware.** We choose the widely used Aloha robot (Fu et al., 2024; Liu et al., 2024a) as our target platform, with three cameras providing multi-view observations. The action for the robot is the target absolute joint position, and does not depend on history. Detailed hardware configurations are in Appendix D.

**Datasets.** For pretraining, we use a curated dataset of one million video clips, sampled from four diverse sources: Egodex (Hoque et al., 2025), Agibot (AgiBot-World-Contributors et al., 2025), RDT (Liu et al., 2024a), and RoboMind (Wu et al., 2024). This large-scale, multi-domain pretraining enables the model to learn rich visual and temporal representations of robotic and human interactions. For finetuning data, finetuning is performed in two domain-specific datasets , including simulation and real-world:

- RoboTwin: We collect 20 episodes for each task on the agilex Aloha platform, resulting in a total of 1,000 episodes.
- Vidarc: We collected 2,307 episodes of high-quality, real-world robot operation data on our Aloha robot platform.

More details of the datasets are shown in Appendix A.

**Baselines.** To ensure fair and meaningful comparisons, we implement two competitive baselines:

- Vidar (Feng et al., 2025): We replicate the Vidar approach using Wan2.2 (Wang et al., 2025) backbone. This baseline undergoes 10k steps of continued pretraining on our pretraining dataset, followed by 14k steps of fine-tuning on each downstream task (RoboTwin and Vidarc), matching our model's fine-tuning budget and origin paper settings.
- Pi0.5 (Intelligence et al., 2025): A strong VLA baseline. Due to architectural and optimization differences, Pi0.5 requires more steps to converge on the relatively scarce data for each task and on extensive tasks. To ensure a fair comparison under the multi-task setting, we fine-tune it over the whole dataset instead.

Our model is built upon the Vidar model that was fine-tuned on the downstream task as a weight warm-up initial, augmented with a teacher-forcing mechanism during training, as detailed in Section 2.1. We fine-tune the model with 4k steps separately on each of the two downstream datasets to adjust the model to capture the ability of causal generation. More training details are listed in Appendix B. All models are evaluated on both the RoboTwin benchmark and our real-world deployment.

## 4.2 MAIN EXPERIMENTS

### 4.2.1 SIMULATION

Average success rates across 14 tasks and success rates for selected tasks are shown in Table 1, where Vidarc achieves high success rates (**C1**). Vidarc is capable of performing complex tasks with remarkable precision, such as grasping a roller using both arms and opening the articulated laptop. Especially for tasks requiring precise bimanual collaborations, such as handing over the microphone, Vidar achieves higher success rates than Vidar, demonstrating the benefits of closed-loop control. Detailed success rates are provided in Table 6, Appendix C.

Table 1: Success rates of different methods and configurations over 14 tasks on the RoboTwin benchmark, tested over 20 episodes. "Average*" means the average of all 14 tasks.

| Method | Average* | Handover Mic | Open Laptop | Place Can Basket | Place Cans Plasticbox |
|---|---|---|---|---|---|
| Pi0.5 | 52.9% | 20.0% | 30.0% | 35.0% | 15.0% |
| Vidar | 71.1% | 0.0% | 50.0% | 50.0% | 0.0% |
| Vidarc | **80.7%** | 65.0% | 55.0% | 45.0% | 85.0% |
| w/o Embodiment-aware | 74.6% | 50.0% | 65.0% | 20.0% | 70.0% |
| w/o Closed-loop | 66.8% | 25.0% | 40.0% | 35.0% | 50.0% |

#### 4.2.2 REAL-WORLD

Real-world experimental results are summarized in Table 2, where Vidarc achieves superior performance over Vidar and Pi0.5 (**C1**). Across three scenarios, Vidarc achieves good generation ability (**C2**) as well as adaptation to environmental changes (the dynamic case) with error correction abilities (**C3**). Visualizations of error correction cases are shown in Figure 4, demonstrating the advantages of our closed-loop control and acceleration methods. Detailed success rates are provided in Table 7, Appendix C.

Table 2: Success rates of different methods over real-world scenarios. "Dynamic" means we manually change the position of the targeted object during execution. Vidarc achieves consistently high success rates across all these scenarios.

| Method | Average | Seen | Unseen | Dynamic |
|---|---|---|---|---|
| Pi0.5 | 41.0% | 48.0% | 28.0% | 48.0% |
| Vidar | 39.0% | 72.0% | 44.0% | 0.0% |
| **Vidarc (Ours)** | **56.0%** | 72.0% | 56.0% | 40.0% |

#### 4.2.3 SPEED EVALUATION

We also conduct a study on the inference speed of various approaches. All experiments are performed under a unified task duration of 6.4 seconds of real-world execution time. We generate 64 frames for video models, as the video fps is 10. For Pi0.5, we generate 16 actions per chunk, 192 actions in total, as the control frequency of Pi0.5 is 30 Hz. All experiments run on a single NVIDIA A100 GPU.

We evaluate performance mainly using two metrics: latency (measured by time to next chunk execution) and end-to-end generation cost (total chunk generation time). As is shown in Table 3, Vidar suffers from high latency due to its large chunk size and the quadratic complexity of its attention operations; consequently, its latency equals its end-to-end cost, which is substantially high. In contrast, Vidarc reduces latency by 91%, mainly benefiting from its causal generation mechanism. With ongoing hardware advances and further model optimizations—such as quantization and distillation—real-time video generation appears increasingly feasible.

Table 3: Inference speed of different methods (in seconds). Vidarc achieves a lower end-to-end cost and a significantly lower latency than Vidar, making great achievements towards the traditionally fast VLA method Pi0.5.

| Method | Latency | Prefill Cost | VAE Cost | Diffusion Cost | End-to-end Cost |
|---|---|---|---|---|---|
| Pi0.5 | 0.482 | - | - | - | 5.76 |
| Vidar | 34.3 | - | 6.25 | 26.9 | 34.3 |
| Vidarc | 3.03 | 0.896 | 6.45 | 10.3 | 24.2 |

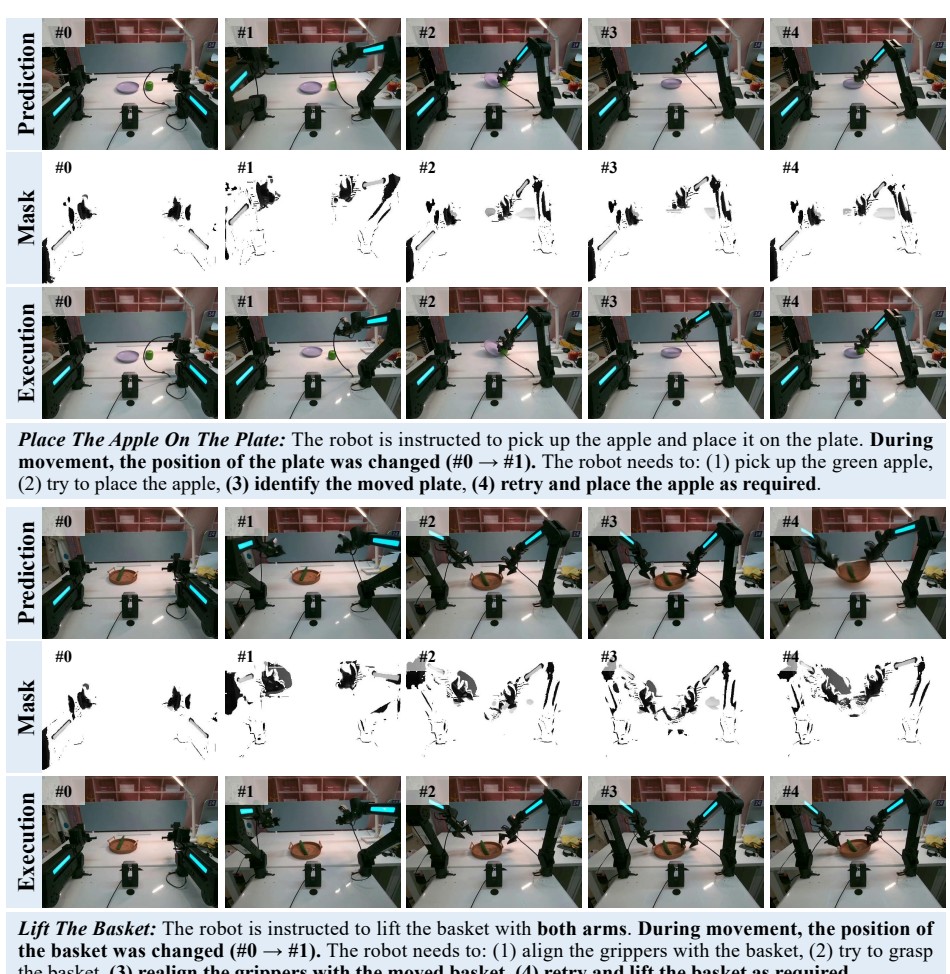

*Place The Apple On The Plate:* The robot is instructed to pick up the apple and place it on the plate. **During movement, the position of the plate was changed (#0 → #1).** The robot needs to: (1) pick up the green apple, (2) try to place the apple, **(3) identify the moved plate**, **(4) retry and place the apple as required**.

*Lift The Basket:* The robot is instructed to lift the basket with **both arms**. **During movement, the position of the basket was changed (#0 → #1).** The robot needs to: (1) align the grippers with the basket, (2) try to grasp the basket, **(3) realign the grippers with the moved basket**, **(4) retry and lift the basket as required**.

Figure 4: Video predictions, corresponding masks, and executions of Vidarc for dynamic tasks, where its error correction ability is observed.

## 4.3 ABLATION STUDY

We conduct ablation studies on the RoboTwin benchmark to systematically evaluate the contributions of (1) the embodiment-aware diffusion loss and (2) closed-loop control enabled by real-world prefilling. As is shown in Table 1, removing embodiment-aware diffusion loss or closed-loop control lowers success rates, which provides solid evidence for **C4**. Detailed results are in Appendix C, where we also conduct a sensitivity analysis of the hyperparameter $\eta$.

## 4.4 CASE STUDY

As illustrated in Figure 5 and Figure 6, a case study is presented to demonstrate how the re-prefilling mechanism effectively bridges the gap between prediction and execution, thereby enhancing model performance. The upper image sequence was execution environment, the under image was the generated frames of the model.

## 5 RELATED WORK

**Vision-Language-Action Methods for Robotics.** Vision-Language-Action (VLA) models leverage natural language instructions as task conditions, enabling multi-task manipulation capabilities

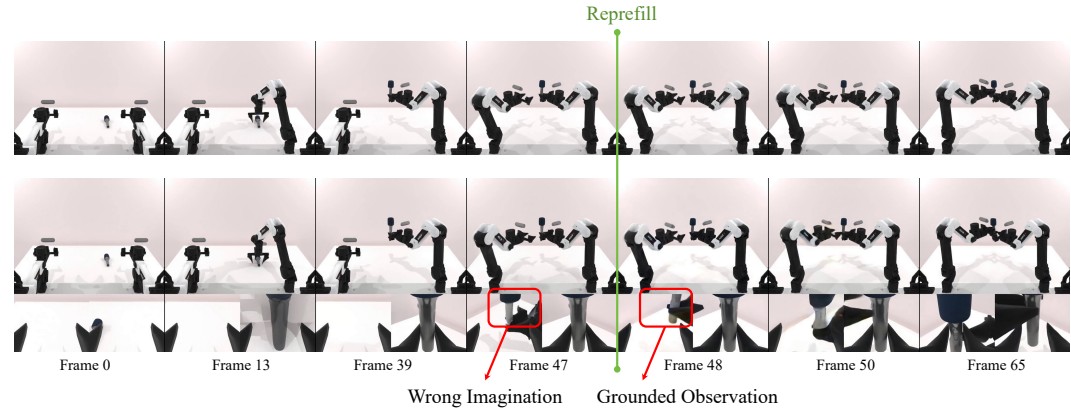

Figure 5: Execution of the method with closed-loop feedback. At frame 47, the model was grounded with real-world sensory data to correct generative drift, ensuring successful task execution.

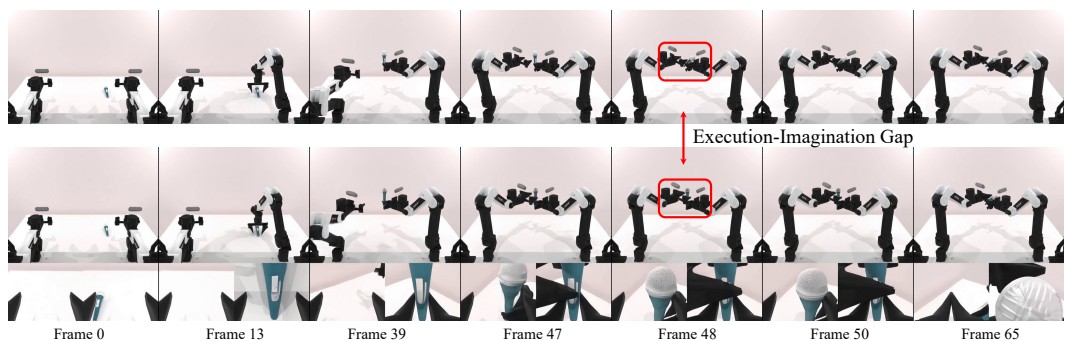

Figure 6: Execution of the method without closed-loop feedback. The model accumulates compounding errors due to ungrounded imagination, leading to eventual task failure.

that go beyond traditional embodied policies such as Diffusion Policy (Chi et al., 2023), which are typically restricted to single tasks. However, the main limitation of current VLA methods is their dependence on enormous, task-conditioned datasets—often comprising thousands of trajectories. The scarcity of such large-scale, richly annotated data severely curtails the broader application of VLA models. Recent advances, including OpenVLA (Kim et al., 2024), Pi0 (Black et al., 2024), Pi0.5 (Intelligence et al., 2025), and RDT-1B (Liu et al., 2024a), have relied on millions of real robot demonstrations spanning diverse embodiments. Despite the considerable scale of these datasets, VLA models still struggle to generalize robustly to unseen tasks or novel environments. Thus, there remains an urgent need for approaches that are both more data-efficient and more generalizable.

**Video World Models for Robotics.** Numerous studies have explored using video world models to decouple image and action spaces. Early approaches (Ha & Schmidhuber, 2018; Schmidhuber, 2015; 1990) utilized RNN-based models and controller architectures to encode visual information and decode actions, respectively. Building on this, recent methods have further explored video-action decoupling, primarily leveraging text-conditioned video generation (Du et al., 2023; Zhou et al., 2024; Bharadhwaj et al., 2024), with extensions including long-horizon planning (Du et al., 2024), 3D data utilization (Zhen et al., 2025), diverse datasets (Yang et al., 2024), and joint video-action latent spaces (Li et al., 2025). Despite all these advances, these methods still suffer from physical inaccuracies, kinematic collapses, and susceptibility to background distractions, especially when confronted with out-of-domain observations. To mitigate these limitations, subsequent work such as Vidar (Feng et al., 2025) extends this paradigm by introducing a two-stage framework for video generation model training and a masked IDM that ignores visual distractors to focus on the robot's arms, thereby enhancing generalization to both novel tasks and backgrounds. However, it still exhibits limited video-level controllability and significant computational overhead. In parallel,

efficient approaches such as Vidman (Wen et al., 2024) and VPP (Hu et al., 2024) have emerged, prioritizing efficiency over the video-action decoupling principle. However, this design choice limits their capability, as they do not model tasks entirely within the visual observation space. This underscores the need for a video world model that simultaneously ensures physical accuracy, optimizes computational cost, and operates within the visual observation space.

**Autoregressive Video Diffusion Models.** State-of-the-art video generation methods, especially those based on diffusion models, have achieved remarkable progress in the quality and temporal consistency of synthesized content (Bao et al., 2024; Wang et al., 2025; Gao et al., 2025b). Inspired by the success of autoregressive frameworks in language modeling, recent studies have increasingly applied autoregressive strategies to video synthesis. Here, pre-trained text-to-video diffusion models generate future frames sequentially, conditioning on previously generated content—examples include NOVA (Deng et al., 2025), NFD (Cheng et al., 2025), Self-Forcing (Huang et al., 2025), Diffusion-Forcing (Chen et al., 2024), FAR (Gu et al., 2025), MAGI-1 (Teng et al., 2025), and CausVid (Yin et al., 2025). This framework also supports interactive video generation, as demonstrated by Matrix-Game 2.0 (He et al., 2025b). However, a primary challenge remains: the substantial inference latency introduced by the iterative diffusion denoising process. To address this, key-value (KV) caching is widely adopted to accelerate decoding during inference (Deng et al., 2025; He et al., 2025b; Huang et al., 2025; Yin et al., 2025; Teng et al., 2025; Cheng et al., 2025). Since the timesteps of previous frames are fixed, features from denoised chunks can be cached and efficiently reused for subsequent frames, eliminating redundant computations and significantly enhancing inference efficiency.

## 6 Conclusion

In this work, we presented Vidarc, a novel framework that integrates an autoregressive embodied video diffusion model with a masked inverse dynamics model to address the challenges of fast, precise control and generalization in data-limited embodied agent settings. By leveraging environmental feedback for closed-loop control, adopting KV cache acceleration, and introducing an embodiment-aware diffusion loss that highlights action-relevant regions, Vidarc overcomes key limitations of existing video-based methods.

Extensive experiments demonstrate that Vidarc achieves significantly higher task success rates, lower latency, and superior generalization to unseen platforms and environments, while also providing robust error correction in real time. Our results highlight the potential of video-based closed-loop methods for scalable and adaptable embodied intelligence, and we believe Vidarc establishes a new direction for efficient and transferable robot learning in complex, dynamic environments.

## 7 Ethics Statement

Vidarc offers the potential to develop generalist, low-latency robot policies with built-in error correction capabilities for real-world environments. However, deploying such systems in sensitive or home settings may also introduce important safety and privacy concerns.

## 8 Reproducibility Statement

We include our code in the supplemental materials, covering both the video diffusion model and the masked inverse dynamics model. To support reproducibility, we also plan to open-source our code and model checkpoints. Appendix A provides a detailed description of our dataset, noting that all pre-training datasets are publicly available. Additional information on training and inference procedures is presented in Appendix B.

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

Table 4: Detailed information about pre-training and fine-tuning datasets.

| Dataset | Size | Type | Camera |
|---|---|---|---|
| Egodex | 230,949 | Human | a movable front camera |
| Agibot | 728,209 | Genie-1 Robot | a fixed high camera, a movable left arm camera, and a movable right arm camera |
| RDT | 6,083 | Aloha Robot | a fixed front camera, a movable left arm camera, and a movable right arm camera |
| RoboMind Franka | 9,589 | Franka Robot | a fixed camera on the opposite side, a fixed left camera, and a fixed right camera |
| RoboMind Aloha | 7,272 | Aloha Robot | a fixed front camera, a movable left arm camera, and a movable right arm camera |
| RoboTwin | 1,000 | Aloha Robot | a fixed rear camera, a movable left arm camera, and a movable right arm camera |
| Vidarc | 2,307 | Aloha Robot | a fixed rear camera, a movable left arm camera, and a movable right arm camera |

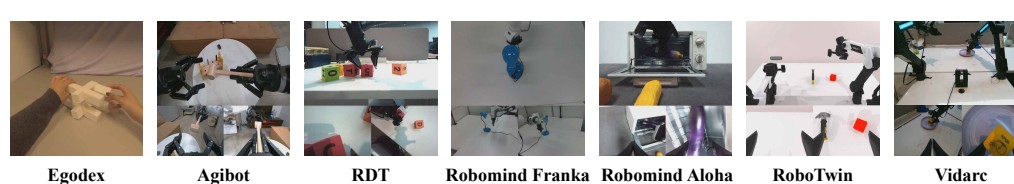

**Egodex**  **Agibot**  **RDT**  **Robomind Franka**  **Robomind Aloha**  **RoboTwin**  **Vidarc**

Figure 7: Visualizations of datasets.

## A  DATASET DETAILS

Detailed dataset information is shown in Table 4. For pre-training, we include human manipulation videos as well as bimanual manipulation videos from 3 different embodiments and various camera configurations. Notably, all these datasets are publicly available. For fine-tuning, we collect 1,000 episodes across 50 tasks on the RoboTwin benchmark; we also collect 2,307 episodes across 219 tasks on our target real-world platform. Notably, the camera and robotic arms for the fine-tuning datasets are unseen and totally different from pre-training. We also provide visualizations of datasets in Figure 7.

We adopt the unified observation space (Feng et al., 2025) for the formatting of all the datasets, which forms a consistent resolution of 720×640.

## B  MODEL, TRAINING, AND INFERENCE DETAILS

### B.1  TRAINING

The complete set of training hyperparameters is provided in Table 5.

For video-based models, input videos are downsampled to 10 frames per second (fps) and resized to a resolution of 736×640. To support classifier-free guidance, the text conditioning is replaced with an empty string with a probability of 0.1 during training.

Our implementation of Vidar adopts the Wan2.2 backbone. The training proceeds in two stages: (i) 10,000 pretraining steps on our internal pretraining dataset, followed by (ii) 14,000 fine-tuning steps on each downstream task (RoboTwin and Vidarc). This two-stage training consumes approximately 4,500 A100 GPU hours in total.

Vidarc is initialized from the reproduced Vidar model and further fine-tuned for an additional 4,000 steps on each downstream task. Notably, the Inverse Dynamics Model (IDM) is shared between

Table 5: Hyperparameters for the training of our experiments.

| Hyperparameter | Vidar/Vidarc | IDM | Pi0.5 |
|---|---|---|---|
| Number of Parameters | 5 Billion | 92 Million | 2Billion |
| Learning Rate | $2 \times 10^{-5}$ | $5 \times 10^{-4}$ | $2.5 \times 10^{-5}$ |
| Batch size | 128 | 128 | 32 |
| Warm-up | 200 Steps | 6k steps | 1k steps |
| Optimizer | AdamW | AdamW | AdamW |
| AdamW $\beta$ | $(0.9, 0.999)$ | $(0.9, 0.999)$ | $(0.9, 0.95)$ |
| AdamW $\epsilon$ | $10^{-8}$ | $10^{-8}$ | $10^{-8}$ |
| AdamW Weight Decay | 0.1 | $10^{-2}$ | $10^{-10}$ |

Vidar and Vidarc. The IDM is trained separately for 60,000 steps with a weighting coefficient $\lambda = 3 \times 10^{-3}$.

For Pi0.5, due to architectural and optimization differences that lead to slower convergence, we fine-tune the model for 45,000 steps on the simulation dataset and 55,000 steps on the real-world dataset. To ensure a fair comparison across methods, we do not perform task-specific fine-tuning within individual tasks of the dataset (i.e., all tasks within a dataset share the same fine-tuned checkpoint).

### B.2 INFERENCE

For the RoboTwin benchmark, we choose 20 sampling steps for both Vidarc and Vidar; the chunk size of Vidarc is set as 16.

For the real-world test, we chose 5 sampling steps for Vidarc and 15 steps for Vidar. In the dynamic scene, we choose a chunk size of 12 for Vidarc and 16 for all other settings.

In speed tests, Vidarc uses 5 sampling steps and generates 8 frames per chunk, whereas the Vidar variant utilizes 20 sampling steps. With our chunk re-prefill method, Vidarc gains an extra 6% end-to-end speed up compared with fully prefill (from 25.8s to 24.2s).

## C ADDITIONAL RESULTS

The complete versions of our simulation and real-world experiments are shown in Table 6 and Table 7.

Detailed ablations are shown in Table 8 and Table 9. We can see that our modules do contribute to the success rates, while the success rates remain high for a wide range of hyperparameter $\eta$.

## D HARDWARE DETAILS

Hardware details of our robot are shown in Figure 8 and Table 10.

## E USAGE OF LARGE LANGUAGE MODELS

We use large language models to aid and polish writing. We draft the content ourselves and then use large language models to refine it, making the writing clearer, more structured, and easier to understand.

Table 6: Average success rates of different methods over the RoboTwin 2.0 benchmark. Vidarc achieves higher average success rates across these methods.

| Task | Pi0.5 | Vidar | Vidarc |
|------|-------|-------|--------|
| Click Alarmclock | 70.0% | 100.0% | 100.0% |
| Click Bell | 70.0% | 95.0% | 100.0% |
| Grab Roller | 75.0% | 100.0% | 95.0% |
| Handover Mic | 20.0% | 0.0% | 65.0% |
| Lift Pot | 35.0% | 90.0% | 80.0% |
| Open Laptop | 30.0% | 50.0% | 55.0% |
| Place A2B Left | 10.0% | 45.0% | 35.0% |
| Place Burger Fries | 75.0% | 80.0% | 80.0% |
| Place Can Basket | 35.0% | 50.0% | 45.0% |
| Place Cans Plasticbox | 15.0% | 0.0% | 85.0% |
| Press Stapler | 60.0% | 90.0% | 100.0% |
| Shake Bottle | 100.0% | 100.0% | 100.0% |
| Shake Bottle Horizontally | 80.0% | 100.0% | 100.0% |
| Stack Bowls Two | 65.0% | 95.0% | 90.0% |
| Average | 52.9% | 71.1% | **80.7%** |

Table 7: Real-world evaluations for different methods under three scenarios. The notations "L", "R", and "B" denote the left arm, right arm, and both arms, respectively. Vidarc achieves consistently high average success rates across these scenarios.

| Seen | Vidarc | Vidar | Pi0.5 |
|------|--------|-------|-------|
| Dump the Waste Paper - R | 60.0% | 100.0% | 40.0% |
| Dump the Waste Paper - L | 80.0% | 80.0% | 100.0% |
| Grasp the Radish - L | 100.0% | 80.0% | 60.0% |
| Wipe Table - L | 60.0% | 40.0% | 20.0% |
| Lift the Basket - B | 60.0% | 60.0% | 20.0% |
| Average | 72.0% | 72.0% | 48.0% |
| Unseen | Vidarc | Vidar | Pi0.5 |
| Place the Eggplant - L | 40.0% | 80.0% | 20.0% |
| Place the Cube - L | 60.0% | 40.0% | 60.0% |
| Place the Cube - R | 40.0% | 0.0% | 20.0% |
| Tap Number One - L | 100.0% | 100.0% | 0.0% |
| Place Steel Wool - R | 40.0% | 0.0% | 40.0% |
| Average | 56.0% | 44.0% | 28.0% |
| Dynamic | Vidarc | Vidar | Pi0.5 |
| Dump the Waste Paper - R | 20.0% | 0.0% | 20.0% |
| Dump the Waste Paper - L | 20.0% | 0.0% | 40.0% |
| Place the Yellow Item - L | 60.0% | 0.0% | 80.0% |
| Lift the Basket - B | 60.0% | 0.0% | 0.0% |
| Place the Apple - R | 40.0% | 0.0% | 100.0% |
| Average | 40.0% | 0.0% | 48.0% |
| All Average | **56.0%** | 39.0% | 41.0% |

Table 8: Average success rates of different configurations over the RoboTwin 2.0 benchmark. "w/o Embodiment-aware" means using the vanilla diffusion loss, and "w/o Closed-loop" means inferring 60 frames using one environmental frame each time.

| Task | Vidarc | w/o Embodiment-aware | w/o Closed-loop |
|---|---|---|---|
| Click Alarmclock | 100.0% | 100.0% | 100.0% |
| Click Bell | 100.0% | 100.0% | 100.0% |
| Grab Roller | 95.0% | 75.0% | 95.0% |
| Handover Mic | 65.0% | 50.0% | 25.0% |
| Lift Pot | 80.0% | 75.0% | 55.0% |
| Open Laptop | 55.0% | 65.0% | 40.0% |
| Place A2B Left | 35.0% | 35.0% | 35.0% |
| Place Burger Fries | 80.0% | 80.0% | 55.0% |
| Place Can Basket | 45.0% | 20.0% | 35.0% |
| Place Cans Plasticbox | 85.0% | 70.0% | 50.0% |
| Press Stapler | 100.0% | 95.0% | 100.0% |
| Shake Bottle | 100.0% | 100.0% | 95.0% |
| Shake Bottle Horizontally | 100.0% | 100.0% | 95.0% |
| Stack Bowls Two | 90.0% | 80.0% | 55.0% |
| Average | **80.7%** | 74.6% | 66.8% |

Table 9: Average success rates of different configurations over the RoboTwin 2.0 benchmark. $\eta$ is the weight hyperparameter in the embodiment-aware loss, and $\eta = 0$ case degrades to the vanilla diffusion loss.

| Task | Vidarc ($\eta = 0$) | Vidarc ($\eta = 3$) | Vidarc ($\eta = 10$) |
|---|---|---|---|
| Click Alarmclock | 100.0% | 100.0% | 100.0% |
| Click Bell | 100.0% | 100.0% | 100.0% |
| Grab Roller | 75.0% | 95.0% | 85.0% |
| Handover Mic | 50.0% | 65.0% | 50.0% |
| Lift Pot | 75.0% | 80.0% | 65.0% |
| Open Laptop | 65.0% | 55.0% | 65.0% |
| Place A2B Left | 35.0% | 35.0% | 30.0% |
| Place Burger Fries | 80.0% | 80.0% | 85.0% |
| Place Can Basket | 20.0% | 45.0% | 35.0% |
| Place Cans Plasticbox | 70.0% | 85.0% | 85.0% |
| Press Stapler | 95.0% | 100.0% | 95.0% |
| Shake Bottle | 100.0% | 100.0% | 100.0% |
| Shake Bottle Horizontally | 100.0% | 100.0% | 100.0% |
| Stack Bowls Two | 80.0% | 90.0% | 85.0% |
| Average | 74.6% | **80.7%** | 77.1% |

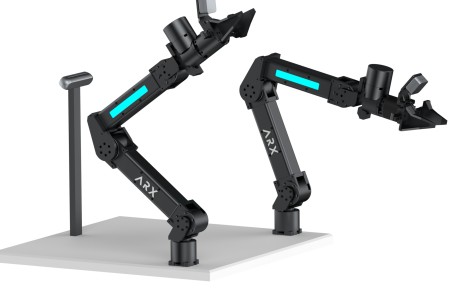

Figure 8: Our robotic platform.

Table 10: Hardware Information.

| Parameters | Values |
|---|---|
| Cameras | 3 RGB Cameras |
| Degree of Freedom | 14 |
| Arm weight | 3.9 kg |
| Arm Valid Payload | 1.0 kg |
| Arm Reach | 0.6 m |
| Arm Repeatability | 1 mm |
| Gripper Range | 0 - 80 mm |
| Gripper Max Force | 10 N |

