# OpenReview forum: "Vidarc: Low Latency Embodied Video Diffusion Model with Closed-loop Control"
_ICLR.cc/2026/Conference — Submitted to ICLR 2026_

### Official Review · Reviewer_tgFC · 2025-10-31

**Soundness:** 2
**Presentation:** 3
**Contribution:** 2
**Rating:** 4
**Confidence:** 4

**Summary:**

This paper proposes VidARC, a framework for low-latency action reconstruction in embodied vision-language-action (VLA) models. The authors address the common inefficiency of current VLA agents, which process perception and action sequentially and suffer from high inference latency during video-conditioned control. VidARC introduces an Action Reconstruction Transformer that incrementally predicts future action embeddings conditioned on video context and recent actions, enabling chunk-wise autoregressive decoding. The system can thus execute actions continuously while maintaining visual grounding. The framework is evaluated in both simulated and real-world environments, showing significant improvements in latency and control stability compared to existing diffusion-based and autoregressive policies.

**Strengths:**

The paper is clearly organized with informative figures and step-by-step reasoning that make the approach easy to follow.

The focus on low-latency action reconstruction fills a clear gap between pure video understanding and real-time embodied control.

The chunk-wise autoregressive inference mechanism is elegant and practically motivated. It effectively balances context utilization and temporal efficiency.

**Weaknesses:**

The paradigm used in this paper has already been extensively explored in prior works such as UniPi[1], VLP[2], RoboDreamer[3], UniSim[4], MPI[5], WorldSimBench[6], TesserAct[7], and Unified Video Action Model[8]. These studies have discussed the potential and applications of the video policy framework from different perspectives, including planning, execution, and benchmarking. The lack of discussion and citation of these related works makes it difficult to fully assess how this paper differs from and contributes beyond them.

I acknowledge that accelerating video policies is an important research direction. However, the introduction of the KV-cache mechanism appears to be more of an engineering implementation rather than a conceptual or methodological innovation. Moreover, the video autoregressive generation component seems closely related to teacher forcing and diffusion forcing, which may further limit the paper’s technical novelty.

Regarding the mask mechanism, I have some concerns. If the mask generation performs poorly in out-of-distribution (OOD) scenarios, would it significantly affect the performance of the video policy compared to approaches without a mask? In addition, since mask training requires additional data annotation, does this process restrict the scalability of the framework? Alternatively, if automated labeling is used, could it introduce error accumulation over time (for example, when the ground-truth masks are inaccurate)?

I would also like the authors to clarify how the mask converges to focus on the robotic arm regions, as shown in the visualizations provided in the paper.

As for the experiments, I believe that using only RobotWin as the simulation benchmark is not sufficiently comprehensive. Many existing video policy studies mentioned above typically evaluate on CALVIN[9] or LIBERO[10]. Since the authors did not reproduce multiple baselines on RobotWin, adopting more general benchmarks or testing against additional video policy or VLA baselines could improve the credibility and completeness of the experimental results.


[1] UniPi: Learning universal policies via text-guided video generation (NeurIPS 2023)

[2] Video Language Planning (NeurIPS 2023)

[3] RoboDreamer: Learning Compositional World Models for Robot Imagination (ICML2024)

[4] Learning Interactive Real-World Simulators (ICLR2024)

[5] Learning Manipulation by Predicting Interaction (RSS2024)

[6] WorldSimBench: Towards Video Generation Models as World Simulators (ICML2025)

[7] TesserAct: Learning 4D Embodied World Models (ICCV2025)

[8] Unified Video Action Model (RSS2025)

[9] CALVIN: A Benchmark for Language-Conditioned Policy Learning for Long-Horizon Robot Manipulation Tasks (RAL)

[10] LIBERO: Benchmarking Knowledge Transfer for Lifelong Robot Learning

**Questions:**

See Weakness.

---

> ### Author Response · Authors · 2025-11-18
> **Reply to Reviewer tgFC (part1)**
>
> We sincerely thank the reviewer for their detailed feedback and for recognizing our paper's clarity, motivation, and elegant design. To prevent any ambiguity and better frame our responses, we would first like to gently clarify a point from the summary. Our Vidarc framework is a **causal autoregressive video diffusion model** that predicts **future visual observations** based on past observations. Actions are then decoded from these predicted video frames using a separate inverse dynamics model. This distinction is central to our contribution. We will ensure this is more prominent in the revised manuscript.
>
> We appreciate the opportunity to address the concerns raised:
>
> ### **1. Regarding Novelty and Relation to Prior Work (UniPi, VLP, etc.)**
>
> We thank the reviewer for pointing out several important related works. We will update our manuscript to include a detailed discussion of these papers and better contextualize our contributions. While we acknowledge the shared paradigm of combining video prediction with an inverse dynamics model, our work makes a distinct and critical contribution by focusing on **low-latency closed-loop control**, a fundamental challenge that prior works like Vidar and UniPi suffer from and that is essential for real-world robotic interaction.
>
> Our key differentiators are:
> *   **Focus on Closed-Loop Interaction:** Unlike many prior works that focus on open-loop planning or offline generation, our core technical innovation is the design of a system that enables high-frequency, real-time feedback integration. For instance, VLP, while powerful in planning, explicitly notes high latency (30 minutes for planning) and hallucination issues as limitations, which our re-prefilling mechanism is specifically designed to address by grounding the model in real-time observations.
> *   **Solving the Latency Barrier:** While acceleration is an important research direction, the challenge for video-based policies is not just an engineering problem but a fundamental system design issue. We address this by building our framework on a causal, autoregressive foundation. This is a deliberate design choice that enables the synergistic use of KV-caching and our novel **re-prefilling algorithm**. A non-causal, bidirectional model like Vidar or UniPi cannot leverage these techniques for incremental feedback, as they would require costly re-computation over the entire history for each step. Our approach reduces latency by 91% compared to Vidar, a crucial step towards practicality.
> *   **Orthogonal Contributions:** Many of the cited works have orthogonal goals. For example, RoboDreamer focuses on compositional generalization, MPI on visual representation learning, and TesserAct on the benefits of 3D data. Our low-latency closed-loop framework is complementary to these advances and can potentially be integrated with them to create even more powerful systems.
>
> In summary, while we build upon established components, our novelty lies in the **holistic system design** that synergistically integrates causal generation, KV-caching, and a bespoke re-prefilling algorithm to solve the critical, previously unaddressed challenge of enabling practical, low-latency, and reactive control for large-scale video policies.

---

> ### Author Response · Authors · 2025-11-18
> **Reply to Reviewer tgFC (part2)**
>
> ### **2. Regarding the Mask Mechanism (OOD Performance and Data Annotation)**
>
> We appreciate the reviewer's concerns and would like to clarify three key points about our mask mechanism:
>
> *   **No Additional Annotation Required:** The masked inverse dynamics model (MIDM) is trained **end-to-end** using only (image, action) pairs from robot demonstrations. It does **not** require any manual mask annotations. The mask is learned implicitly.
> *   **How the Mask Learns:** The mask converges to focus on action-relevant regions (like the robot arm) through a clever interplay of two signals during training:
>     1.  **Action Prediction as Supervision:** The primary learning signal comes from the action prediction error. If the mask incorrectly covers the background or misses a part of the arm, the action prediction will be inaccurate. The backpropagated error forces the mask predictor to focus only on regions that are informative for predicting the ground-truth action.
>     2.  **L1 Sparsity Regularization:** A simple L1 penalty is applied to the mask's area. This encourages the mask to be as sparse as possible while still enabling accurate action prediction, effectively forcing it to discard irrelevant background pixels.
> *   **Decoupling and Robustness:** During inference, the video model's generation and the MIDM's mask prediction are **decoupled**. The video model first generates a future frame $\hat{o _ t}$. Then, the separately trained MIDM predicts a mask and action from that frame $\hat{o _ t}$. The mask's role in the video model is as a **training-time regularizer** (the embodiment-aware loss), guiding the model to improve generation quality in critical areas. It does not directly participate in the video generation loop at inference time. Therefore, an OOD scenario for the mask predictor would not cause a compounding failure in the video generation process itself. The robustness of this MIDM approach, even in unseen environments, is one of the key contributions of Vidar, which our work builds upon.
>
> We have clarified these details in the revised manuscript to make the mechanism more transparent.
>
> ### **3. Regarding Experimental Benchmarks**
>
> We agree with the reviewer that evaluation on multiple standard benchmarks is crucial for comprehensive validation. The RoboTwin benchmark was initially chosen as it provides a standardized evaluation platform with a public leaderboard, where the performance of many strong baselines is already officially reported. This allows for a direct and fair comparison against a wide range of existing methods. To further strengthen our evaluation, we are in the process of running experiments on the **LIBERO benchmark** as suggested and will include these results in the final version of the paper.
>
> We thank the reviewer again for their constructive and valuable feedback, which has helped us significantly improve the clarity and completeness of our work.

---

### Official Review · Reviewer_k5Kh · 2025-10-31

**Soundness:** 3
**Presentation:** 3
**Contribution:** 3
**Rating:** 4
**Confidence:** 3

**Summary:**

This paper proposes Vidarc, an autoregressive embodied video diffusion framework integrated with a masked inverse dynamics model, designed to address the challenges of low-latency closed-loop control and generalization in data-scarce robotic manipulation scenarios. By introducing an embodiment-aware diffusion loss guided by action-relevant masks and incorporating KV caching for real-time environmental feedback, Vidarc aims to enhance the alignment between video generation and robotic embodiment dynamics. Pre-trained on one million cross-embodiment episodes and fine-tuned on unseen platforms, the model demonstrates higher task success rates (15-17% higher than baselines like Vidar and Pi0.5) and lower latency in both simulated and real-world experiments, along with error correction capabilities.

**Strengths:**

The integration of autoregressive video diffusion with closed-loop control via KV caching and environmental feedback re-prefilling addresses the high-latency issue of traditional non-autoregressive video-based methods, enabling more responsive robotic manipulation.
The embodiment-aware loss, weighted by masks from the inverse dynamics model, effectively prioritizes action-relevant regions, mitigating the problem of irrelevant visual distractions and improving the actionable quality of generated videos.
Extensive pre-training on large-scale cross-embodiment datasets ensures strong generalization to unseen robotic platforms and tasks, with experimental results validating superior performance over state-of-the-art baselines in both simulation and real-world settings.

**Weaknesses:**

Unclear Visualization and Ambiguous Definitions: The error correction example in Figure 1 lacks sufficient explanation, making it difficult to understand the model’s error correction mechanism. Additionally, the definition of the observation space O is ambiguous—if O is a single image, predicting actions from a single frame contradicts the fundamental definition of inverse dynamics models and poses an ill-posed problem.
Incomplete Literature Review: The paper claims that existing video-based methods are inefficient but overlooks prominent efficient approaches such as Vidman and VPP, leading to an incomplete assessment of the current research landscape.
Practical Efficiency Limitations: Despite adopting CausVid for optimization, Vidarc’s reliance on the Wan2.2 backbone results in excessive memory overhead during training and inference. With a per-step latency of 3 seconds, the model fails to meet the real-time requirements of embodied intelligence applications.
Thin Simulation Experiments: The simulation evaluations are only conducted on the RoboTwin benchmark, lacking validation on other widely used benchmarks like Libero or RLBench. This limits the generalizability of the reported performance.
Ambiguous Contribution of Core Components: The performance advantage of Vidarc over baselines is primarily attributed to closed-loop control, but this component is not inherently tied to video generation—any autoregressive method could integrate such feedback. Moreover, removing closed-loop control leads to a significant performance drop (66.8% vs. 80.7% average success rate on RoboTwin), casting doubt on the necessity of the embodiment-aware loss. The weighted loss may also neglect background information critical for diffusion denoising, yet no theoretical justification or quantitative analysis of the loss’s feasibility is provided.
Insufficient Analysis of Key Mechanisms: The masked inverse dynamics model uses different input sources during training and inference, raising concerns about the stability and controllability of its performance, especially with observations generated via embodiment-aware training. Additionally, there is a lack of case studies and quantitative analysis on how closed-loop control improves action accuracy over iterations, leaving the mechanism’s effectiveness underexplored.

**Questions:**

Overall, the concept of Vidarc is relatively straightforward, and many implementation details lack sufficient empirical or theoretical support. The insufficient experimental design raises concerns about whether the reported results are robust or merely coincidental, reducing the work’s ability to provide meaningful guidance for future research.

---

> ### Author Response · Authors · 2025-11-18
> **Reply to Reviewer k5Kh (part1)**
>
> We sincerely thank the reviewer for the thorough and constructive feedback. The reviewer's detailed comments have been invaluable in helping us refine our manuscript. We are pleased that the reviewer recognized the strengths of our work in addressing latency through autoregressive closed-loop control and enhancing actionability with our embodiment-aware loss. We address the identified weaknesses below:
>
> ### **1. On Unclear Visualization and Ambiguous Definitions**
>
> We thank the reviewer for pointing out the lack of clarity.
>
> - **Error Correction Mechanism:** Our error correction mechanism is enabled by the closed-loop control with re-prefilling. When a deviation occurs between the predicted plan and the actual execution (e.g., due to dynamic environmental changes), our model receives the latest ground-truth observation from the environment. By re-prefilling this true state into the KV cache, the model becomes conditioned on what actually happened, allowing it to perceive the error and generate a new, corrected plan for the subsequent steps. This is a key advantage of our causal, closed-loop approach over open-loop methods. We will revise Figure 1 to better illustrate this mechanism and add a more detailed case study to the Appendix.
>
> - **Observation Space $\mathcal{O}$:** We apologize for the ambiguity in our notation and thank the reviewer for pointing it out. The observation space $\mathcal{O}$ represents a sequence of images. Fundamentally, our Inverse Dynamics Model (IDM) is defined to map a single target observation (an image) to a corresponding action (absolute joint positions). However, in our implementation, we use a chunk-wise inference strategy where the video model generates a sequence of future frames. Consequently, we apply the IDM to this batch of frames to infer a corresponding batch of actions. With a slight abuse of notation, we refer to this batched inference process also as the IDM, $I$. This notation is consistent with the formulation in Vidar. We will clarify this in the revised manuscript to avoid confusion.
>
> ### **2. On Incomplete Literature Review**
>
> We appreciate the reviewer for bringing these prominent works to our attention. We have read Vidman and VPP and agree they are relevant.
>
> - **Vidman** first pre-trains a video model (OpenSora) and then adapts it into an inverse dynamics model. Its formulation is more akin to a diffusion policy that conditions on past frames to predict actions.
>
> - **VPP** uses the latent features from a single denoising step of a video model to augment a separate action model, but it does not fully leverage the video model's generative capabilities for planning. Furthermore, as shown in the original Vidar paper, VPP's performance is lower than Vidar, which is a direct baseline in our work.
>
> - Crucially, both methods differ from our approach as they do not model the task entirely within the visual observation space, which we believe is key for effective knowledge transfer across different embodiments. Our primary contribution remains distinct: enabling low-latency, closed-loop control for video-based policies that plan in observation space. We will add a discussion of these methods to our related work section.
>
> ### **3. On Practical Efficiency Limitations**
>
> We agree with the reviewer that the current 3-second latency does not yet meet the strict requirements for real-time control. However, we wish to clarify that the core contribution of this paper is the architectural innovation that **enables** low-latency closed-loop control for video diffusion models, demonstrated by a **10-fold (91%) reduction in latency** compared to the non-autoregressive Vidar baseline. Our focus is on demonstrating the feasibility and benefit of the causal autoregressive framework. Moreover, recent works like Genie 3[1] and Matrix Game 2.0[2] have shown that large causal generative models can achieve real-time performance through established techniques like model distillation and system optimization. These techniques are orthogonal to our core contribution but present a clear and viable path for future work to build upon our foundation and achieve true real-time inference.

---

> ### Author Response · Authors · 2025-11-18
> **Reply to Reviewer k5Kh (part2)**
>
> ### **4. On Thin Simulation Experiments**
> Thank you for this suggestion. We chose the RoboTwin benchmark because it is comprehensive and its platform closely aligns with our real-world hardware setup, allowing for more direct sim-to-real comparison. We acknowledge the value of broader evaluation and, as suggested, we will do our best to conduct additional experiments on the LIBERO benchmark during the rebuttal period and include the results in the final version.
>
> ### **5. On Ambiguous Contribution of Core Components**
> - **Closed-Loop Control:** We appreciate this insightful comment. We agree that closed-loop control is a critically important component, and demonstrating its effectiveness for video-based models is a central claim of our paper. While the *concept* of closed-loop control is general, implementing it *efficiently* on a large, bidirectional video diffusion model like Vidar is challenging and prohibitively slow, as it cannot leverage KV caching. Therefore, our specific design choice of combining **autoregressive video generation with a re-prefilling mechanism** is a novel and crucial contribution that makes low-latency closed-loop control practical for this class of models.
> - **Embodiment-Aware Loss:** We thank the reviewer for the opportunity to clarify the role of the embodiment-aware loss. We want to emphasize that our loss function, $\mathcal{L} \propto \| (1 + \eta \cdot m) \odot \Delta v \|^2$, *re-weights* the loss on action-relevant regions rather than discarding background information. This ensures the model focuses more on generating kinematically plausible robot motions, which is critical for the IDM to decode accurate actions, without neglecting the overall scene context. To provide quantitative evidence, we measured the mean value of the last 10 training steps of the flow matching loss. The value of the flow-matching loss with the embodiment-aware technique (0.058) is comparable to the baseline flow-matching loss (0.054), indicating that our approach improves focus on critical regions without degrading overall generation quality.
>
> ### **6. On Insufficient Analysis of Key Mechanisms**
> - **IDM Train-Inference Gap:** This is an excellent point regarding the potential input distribution shift for the IDM. We address this concern in two ways: First, the robustness of this masked IDM architecture to generated inputs was previously validated in the Vidar paper. Second, and more importantly, our strong end-to-end task success rates demonstrate that the IDM performs effectively in our pipeline. We have also observed that the vast majority of task failures stem from inaccuracies in the video prediction phase, rather than a failure of the IDM to decode a reasonable action from a well-predicted frame.
> - **Case Studies:** We agree that a detailed case study would strengthen the paper. As suggested, we will add a qualitative analysis in Section 4.4, illustrating how closed-loop control helps correct errors over iterations in dynamic scenarios.
>
> Thank you again for your insightful feedback and genuine interest in advancing the field. We recognize the importance of interpretability and providing clear insights. We will incorporate the promised clarifications, discussions, and additional case studies/experiments into our final manuscript to better support our claims and provide more meaningful guidance for future research. To further contribute to the community and promote diverse research directions, we will be open-sourcing our code and models.
>
> [1] DeepMind. (2025). Genie 3: A new frontier for world models. Retrieved from https://deepmind.google/blog/genie-3-a-new-frontier-for-world-models
>
> [2] He, X., Peng, C., Liu, Z., Wang, B., Zhang, Y., Cui, Q., ... & Zhou, Y. (2025). Matrix-game 2.0: An open-source, real-time, and streaming interactive world model.

---

### Official Review · Reviewer_fS78 · 2025-11-02

**Soundness:** 3
**Presentation:** 3
**Contribution:** 3
**Rating:** 6
**Confidence:** 4

**Summary:**

The authors present an autoregressive video diffusion model that augmented with a masked inverse dynamics model.  The authors show that by grounding the video prediction of the video diffusion model with action-relevant masks, they obtain fast and accurate closed-loop control.  The authors claim the learned masks prioritizes the video generation quality of the action-relevant regions. The masked inverse dynamics model is from an earlier paper called Vidar.  The authors show that the model when pre-trained on one million cross-embodiment episodes outperforms baselines in success rates and latency, even on unseen robotic platforms.

**Strengths:**

- The video generation model incorporates an embodiment-aware loss to make the model focus generation on action-relevant regions.

- Paper produces a method for low-latency video generation for closed-loop control.

- Results work on unseen robotic platforms.

**Weaknesses:**

- The key strength of this paper was to enable the video generation to focus generation on action-relevant regions.  However, this is possible due to the prior work on masked inverse dynamics model (from Vidar?).  This makes the novelty of this work weaker.

**Questions:**

- Figure 1, the left hand side should be drawn in a classical feedback control diagram that is popular in robotics rather than having an execution and environment feedback at the top.  A diagram that shows observation o_t leading to o_t^' producing a_t which then results in o_{t+1} that is feedback is better than the current diagram - which is confusing as the bottom o_1, o_2, o_3 feedback is not clear.  (This is my opinion.)  The figure on the right is disconnected from the figure on the left - this can be revised and the caption can be modified to better connect the two subfigures.

- Figure 2 is missing the action a_1.  Typically, o_{t+1} is produced based on a_t that is computed as a function of o_t.  Here, o_1 -> o_2^' -> a_2, which misses a_1.

- Sec. 2.1: v_\theta : V x R X C -> V, here the spaces V and C are not defined.  Is V the space of the video x and C the space of the condition c?

- The diffusion loss in Eq. (1) looks at the difference of the vector field v_\theta and the mean vector field (x_0 - x_1).  v_theta is the vector field at x_t at t, where as (x_0 - x_1) is just the mean vector field.  How does getting these two to match up be sufficient for diffusion?  Can you double check this loss.

- In Eq. (5), why not have a_t = I(o_t) instead of I(\hat o_t) ?  (I understand that ideally o_t and \hat o_t should be the same, but they typically are not due to training loss not being zero.)

---

> ### Author Response · Authors · 2025-11-18
> **Reply to Reviewer fS78 (part1)**
>
> We sincerely thank the reviewer for their positive assessment and insightful feedback on our work. We are encouraged that the reviewer recognized the strengths of our method in enabling low-latency closed-loop control and its effective generalization to unseen platforms. Below, we address the specific weaknesses and questions raised.
>
> ### **1. Novelty**
>
> We appreciate the reviewer's comment regarding the novelty of our work in relation to Vidar. We agree that our work builds upon the masked inverse dynamics model (MIDM) concept. However, we would like to clarify two primary areas where our contribution is fundamentally novel:
>
> 1.  **Our primary novelty lies in the design of a low-latency, closed-loop system for video-based policies.** The core contribution of Vidarc is the integration of **autoregressive video generation with a real-time re-prefilling mechanism**. This architecture is specifically designed to address the critical challenge of high latency and error accumulation in previous video world models. As demonstrated in our experiments (Table 3), this system reduces inference latency by 91% compared to Vidar and enables robust error correction in dynamic scenes where open-loop methods like Vidar fail completely (Table 2). This advancement is crucial for the practical application of video models in real-world robotics.
>
> 2.  **Our use of the mask is conceptually different and novel.** While Vidar uses the mask solely within the inverse dynamics model to decode actions from a given video, we **repurpose this mask to directly guide the video generation process itself.** By incorporating the mask into our **embodiment-aware diffusion loss** (Equation 7), we compel the video diffusion model to prioritize the generation of physically plausible and action-relevant regions of the robot. This creates a powerful synergy between the world model and the controller, ensuring the generated future is not just visually coherent, but also "actionable". Our ablation study (Table 1) confirms that this novel loss significantly contributes to the final task success (80.7% vs. 74.6%).
>
> In summary, while we leverage the MIDM, our key innovations are the architectural design for low-latency closed-loop control and the novel application of the learned mask to fundamentally improve the underlying video generation model.
>
> ### **2. Figure 1 Diagram**
> Thank you for the constructive suggestion. We agree that a diagram more aligned with classical feedback control theory would significantly improve the clarity of our method. In the revised version, we will redraw Figure 1 to explicitly illustrate the $o _ t \rightarrow \hat{o} _ {t+1} \rightarrow a _ t \rightarrow o _ {t+1}$ feedback loop and better connect the overall architecture on the left with the application examples on the right.

---

> ### Author Response · Authors · 2025-11-18
> **Reply to Reviewer fS78 (part2)**
>
> ### **3. Missing $a _ 1$ in Figure 2**
> Thank you for pointing out this notational inconsistency. It is a notation issue; this diagram shows the training pipeline of the model, and we use $a _ 2$ instead of $a _ 1$ to emphasize that $a _ 2$ is the training target qpos in $o _ 2$. We plan to change the notation from $a _ 2$ to $a _ 1$, which corresponds to the traditional notations.
>
> ### **4. Undefined Spaces $V$ and $C$ in Section 2.1**
> Thank you for catching this omission. The reviewer is correct: $V$ represents the space of the video $x$, and $C$ represents the space of the condition $c$. We will add these formal definitions to the paper.
>
> ### **5. Diffusion Loss in Equation (1)**
> This is an excellent question that allows us to clarify an important detail. The objective in Equation (1) is the simplified training loss for **Flow Matching** [1], which is employed by many modern video diffusion models, including our backbone, Wan 2.2. While closely related to score-based diffusion, the Flow Matching framework directly learns the vector field that transforms a noise distribution to the data distribution. We realize our text did not make this clear. We will revise this section to explicitly state that we are using the Flow Matching framework and add the appropriate citations to avoid any confusion with other diffusion formulations like DDPM[2].
>
> ### **6. $I(\hat{o} _ t)$ vs. $I(o _ t)$ in Equation (5)**
> This is a critical point that highlights the core principle of our closed-loop pipeline. The action $a _ t$ must be determined *before* the new ground-truth observation $o _ {t+1}$ is available from the environment. Therefore, it is impossible to use the actual future observation $o _ {t+1}$ to decode the current action. Instead, the policy must rely on a **prediction of the future**.
>
> Our process is as follows: based on the history of real observations $(o _ 1, \dots, o _ t)$, the model first generates a predicted future observation $\hat{o} _ {t+1}$. The inverse dynamics model then decodes the action $a _ t = I(\hat{o} _ {t+1})$ from this prediction. This action $a _ t$ is then executed to cause the transition to the real next state $o _ {t+1}$. We apologize for the notational confusion in the original manuscript. As the reviewer pointed out, there was a typo. We will revise Equation (5) and the surrounding text to make this causal dependency and the indices crystal clear.
>
> We believe addressing these points will significantly improve the clarity and impact of our paper. We thank the reviewer again for their valuable and constructive guidance.
>
> [1] Lipman, Y., Chen, R. T., Ben-Hamu, H., Nickel, M., & Le, M. (2022). Flow Matching for Generative Modeling.
>
> [2] J Ho, A Jain, P Abbeel.(2020). Denoising diffusion probabilistic models

---

### Meta-Review · Area_Chair_9ynY · 2026-01-11

**Summary:**

This paper presents Vidarc, a causal autoregressive video diffusion model for low-latency closed-loop robotic control.
The submission received reviews from three reviewers, with scores of 6, 4, and 4.
The reviewers raised several substantive concerns. On novelty, all three reviewers questioned whether the contribution is sufficiently novel given heavy reliance on the Masked Inverse Dynamics Model from prior work (Vidar).
The KV-caching mechanism, while useful, represents standard practice in autoregressive models rather than methodological innovation.
On empirical validation, the simulation experiments are limited to the RoboTwin benchmark, with no evaluation on established benchmarks such as LIBERO, CALVIN, or RLBench that would enable comparison with the broader literature.
On practical utility, the 3-second per-step latency remains far from real-time requirements for embodied control, limiting the practical impact of the claimed efficiency gains. On technical depth, multiple reviewers noted that the embodiment-aware loss is a relatively straightforward reweighting scheme, and the paper lacks theoretical justification for why this approach should work.

**Reviewer Concerns:**

Concerns addressed:

The authors provided reasonable clarifications on technical details including the observation space definition, the mask learning mechanism, and the Flow Matching formulation. The quantitative comparison of flow-matching loss values (0.058 vs. 0.054) offers some evidence that the embodiment-aware loss does not degrade generation quality.

Concerns not adequately addressed:

The novelty concern remains the most significant issue. The authors argue that their contribution is "system-level design," but the individual components - causal video generation, KV-caching, inverse dynamics models, and attention masking - are all established techniques. The integration, while competent, does not represent a sufficiently novel intellectual contribution for a top venue.
The benchmark limitation is critical and was not resolved. Despite commitments from the authors, no additional experimental results on LIBERO or other standard benchmarks were provided during the rebuttal period. Given that all three reviewers raised this concern, the absence of broader evaluation substantially weakens confidence in the generalization claims. The RoboTwin benchmark alone is insufficient to support the paper's claims about cross-embodiment transfer.

The efficiency limitation was acknowledged but not adequately contextualized. While a 91% reduction from Vidar's latency sounds impressive, the absolute latency of 3 seconds per step remains impractical for real-world deployment. The authors' argument that future distillation techniques could address this gap is speculative and shifts the burden of demonstrating practical utility to hypothetical future work.

**Reviewer Scores:**

Reviewer fS78 (Initial: 6): This reviewer was the most positive and may maintain the score at 6. However, without seeing the promised figure revisions and given the lack of additional experiments, an increase seems unlikely.

Reviewer k5Kh (Initial: 4): The core concerns about thin simulation experiments and ambiguous contribution of components were not fully resolved. The reviewer explicitly noted that "many implementation details lack sufficient empirical or theoretical support" and questioned whether results are "robust or merely coincidental." Without the promised LIBERO experiments, the score is likely to remain at 4 or possibly decrease.

Reviewer tgFC (Initial: 4): This reviewer raised the most extensive concerns about novelty relative to prior video policy work. The authors' response did not provide convincing differentiation from methods like UniPi, VLP, and others. The score is likely to remain at 4.

While the paper addresses a relevant problem and demonstrates some empirical improvements over the Vidar baseline, the contribution does not meet the threshold for acceptance at this venue for several reasons.

The novelty is limited. The paper combines established techniques (causal autoregressive generation, KV-caching, masked inverse dynamics) without introducing new methodological insights. The embodiment-aware loss, presented as a key contribution, is a straightforward reweighting scheme that lacks theoretical grounding. The extensive body of prior work on video-based policies makes it difficult to identify what fundamentally new knowledge this paper contributes.

The empirical validation is insufficient. Reliance on a single simulation benchmark (RoboTwin) is inadequate for supporting claims about generalization and cross-embodiment transfer. The authors' failure to provide promised results on standard benchmarks during the rebuttal period is concerning, particularly given that this was a unanimous concern across reviewers.

The practical impact is unclear. The 3-second latency, while improved relative to Vidar, remains far from the real-time requirements that would make this approach practically useful. The paper's framing around "low-latency" control is therefore somewhat misleading.

Two of three reviewers scored the paper below the acceptance threshold, and the rebuttal did not substantively address the core concerns that motivated those scores. The paper would benefit from broader experimental validation, clearer differentiation from prior work, and demonstration of practical real-time performance before resubmission.

---

### Decision · Program_Chairs · 2026-01-26

Reject